# Environmental Effect Evaluation: A Quantile-Type Path-Modeling Approach

## Hao Cheng

National Academy of Innovation Strategy, China Association for Science and Technology, Beijing 100038, China; chenghao0524@yeah.net

**Abstract:** The environment is a key element that affects many aspects of our society, including the economy, education and talents. In this article, the main purpose is to provide statistical models, algorithms and quantitative evidence regarding environmental effect evaluation (EEE). To accomplish this investigation, I first establish a theoretical EEE model and then apply a quantile-type path-modeling algorithm in the developed EEE model at different quantile levels. In the real-data analysis, this article investigates hypotheses regarding this theoretical EEE model and illustrates the statistical performances of quantile-type path-modeling EEE estimators through bootstraps. The results mainly illustrate that the environment has indispensable impacts on the economy, education and science and technology talent directly and has indirect effects on scientific infrastructure and science and technology output. Compared with the existing classical path-modeling algorithm, quantile-type path-modeling EEE estimators make full use of quantile regression and then overcome the classical exploration of only average effects. Both the quantile-type EEE model and quantile-type path-modeling algorithm capture changes in the relations among constructs and between the constructs and observed variables, and this helps to analyze the entire distribution of the outcome variables in this EEE model.

**Keywords:** environmental effects; quantile regression; path modeling; model assessment

## 1. Introduction

Various experts and researchers have conducted qualitative investigations regarding different kinds environmental topics [1–9]. Cao et al., (2022) performed a study on the priorities for the development of China Certified Emission Reduction (CCER) forest carbon sink projects under the context of carbon neutrality goals [10]. Zhai et al., (2020) applied a structural equation model and analyzed the influencing factors of green transformation in China's manufacturing industry under environmental regulation [11].

Feng et al., (2017) investigated green development performance and its influencing factors from a global perspective [12]. Carmen et al., (2010) researched environmental innovation and environmental performance [13]. Pearce et al., (1990) studied the economics of natural resources and the environment [14]. Pearce et al., (1990) focused on economics and the environment in the third world and investigated its sustainable development [15]. Arntzen (1989) chose Rural Botswana and studied its environmental pressures and adaptations [16]. Schramm et al., (1989) conducted investigations in environmental management and economic development [17].

Different from the existing studies, this article focuses on investigating environmental effects in a qualitative way and establishing comprehensive evaluation models, which have not been well developed. As we know, the environment has impacts on the economy, education, talents and many other aspects, such as scientific infrastructure and science and technology output. Environmental effect evaluation (EEE) becomes increasingly important and complex and needs to comprehensively consider all the above aspects.

This article first defines the environment as a combination of the natural environment status (including energy, water and air) and social environment conditions (such

as biocapacity, pollution and laws) and then evaluates the environment's effects on the economy, education, science and technology talent, scientific infrastructure and science and technology output. It is easily understandable that a well-developed environment provides a solid basis for life and a good atmosphere and supports the development of the economy, education, science and technology talent, scientific infrastructure and science and technology output. Environmental pollution terribly hinders societal development all over the world.

The main contributions of this article are to provide statistical models, algorithms and quantitative evidence regarding EEE based on the proposed theoretical research hypotheses from the perspective of the entire distribution of variables rather than the classical exploration of only average effects. Thus, I chose a quantile-type EEE model and a quantile-type path-modeling algorithm, which is the well-known quantile composite-based path-modeling approach (Davino et al., 2016) [18]. Based on these quantile-type methodologies, this article is able to evaluate the environment from the perspective of quantiles and to highlight changes in the relations among different aspects.

The introduction of the quantile level brings difficulty; the accomplishment of coefficient estimation requires a quantile-type path-modeling algorithm rather than the classical partial least squares algorithm. Currently, both partial least squares (PLS) and its extensions in quantile regression have been investigated by many researchers in various applications (Chatelin et al., 2002; Jarvis et al., 2003; Ciavolino et al., 2013a and 2013b; Esposito et al., 2010; Davino et al., 2016; Davino et al., 2017; Davino et al., 2018; Cheng 2020) [19–27]. Using a quantile-type path-modeling algorithm, the estimated coefficients can be estimated at each quantile level.

The rest of the paper is organized as follows. The theoretical basis of the research is described in Section 2. In this section, I define the environment and its effects, the research hypotheses, the EEE conceptual framework and the EEE indicator measurement in Sections 2.1–2.4, respectively. Section 3 introduces the data-preprocessing methods and data-preparation work and then describes the data through violin plots. Section 4 describes the methodology, including the quantile-type EEE model in Section 4.1 and quantile-type path-modeling algorithm in Section 4.2. Section 5 applies the models and algorithms to real-data analysis. Section 5.1 shows the path and loading coefficient estimations at different quantile levels. Section 5.2 demonstrates statistical performance investigations through raw estimation and bootstraps. Section 5.3 investigates the model assessment and validation of the quantile-type EEE model. Some final discussions are placed in Section 6.

## 2. Theoretical Basis of the Research

### 2.1. Definitions of the Environment and Its Effects

The environment (EN) is a key issue for society, and, in this paper, it refers to a combination of the natural environment's state and social environmental conditions. The environment can be viewed as an important factor that directly affects the economy (EY), education (ED) and science and technology talent (TA) as well as many other aspects, such as scientific infrastructure (SI) and science and technology output (OU). The International Institute for Management Development (IMD) World Competitiveness Yearbook explains the constructs involving the environment and its effects.

The environment can be reflected through ten institutional indicators, including the energy intensity, the paper and cardboard recycling rate, waste water treatment plants, the water consumption intensity, $CO_2$ emissions, renewable energies, the total biocapacity, sustainable development, pollution problems and environmental laws. Both the GDP per capita and government consumption can explain a country's economic status. Government consumption refers to the consumption expenditure of public services provided by government departments for the entire society and the net expenditure of goods and services provided to households free of charge or at a lower price. Many experts have investigated the relationship between government consumption and economic growth and found a tight association between them (Wu et al., 2009) [28].

Education expenditure per capita, English proficiency and the educational system are appropriate elements to cover the definition of education. To investigate the environmental effects based on 61 economies, English proficiency is used an indicator to reflect the education internationalization level. Here, we do not consider enrollment in higher education mainly due to the following reason: the overall plan for deepening the reform of education evaluation in the new era requires that Party committees and governments at all levels shall not use the enrollment index or the enrollment rate of the college entrance examination as the assessment standard, and it is strictly prohibited to publish, publicize or hype the "top scorer" and enrollment rates of the college entrance examination. This paper does not use enrollment in higher education and instead considers the education expenditure per capita, English proficiency and educational system to explain the definition of education.

Science and technology talent can be expressed by the labor force, researchers and scientists and brain drain. For science and technology output, Nobel prizes and patent applications can be used to reflect a country's science and technology achievements. Scientific infrastructure can be reflected through the total expenditure on R&D, scientific research legislation and intellectual property rights.

### 2.2. Research Hypotheses

According to the analysis of the environment and its effects, I propose the following theoretical hypotheses regarding their relations.

**Hypothesis 1.** *The environment positively affects the economy, while the other variables are unchanged (denoted as **H1**). The EEE model uses $\beta_{1,1}(\tau)$ to represent **H1**.*

**Hypothesis 2.** *The economy positively affects science and technology talent, while the environment and its other effects are unchanged (denoted as **H2**). The EEE model uses $\beta_{5,1}(\tau)$ to represent **H2**.*

**Hypothesis 3.** *Science and technology talent has a positive effect on science and technology output, while the environment and its other effects are unchanged (denoted as **H3**). The EEE model uses $\beta_{3,1}(\tau)$ to represent **H3**.*

**Hypothesis 4.** *The economy has a positive impact on scientific infrastructure, while the environment and its other effects are unchanged (denoted as **H4**). The EEE model uses $\beta_{2,1}(\tau)$ to represent **H4**.*

**Hypothesis 5.** *Education positively impacts science and technology talent, while the environment and its other effects are unchanged (denoted as **H5**). The EEE model uses $\beta_{5,3}(\tau)$ to represent **H5**.*

**Hypothesis 6.** *Scientific infrastructure positively affects science and technology talent, while the environment and its other effects are unchanged (denoted as **H6**). The EEE model uses $\beta_{5,4}(\tau)$ to represent **H6**.*

**Hypothesis 7.** *The environment has a positive impact on science and technology talent, while the other variables are unchanged (denoted as **H7**). The EEE model uses $\beta_{5,2}(\tau)$ to represent **H7**.*

**Hypothesis 8.** *The environment positively affects education, while the other variables are unchanged (denoted as **H8**). The EEE model uses $\beta_{4,1}(\tau)$ to represent **H8**.*

**Hypothesis 9.** *Scientific infrastructure has a positive effect on education, while the environment and its other effects are unchanged (denoted as **H9**). The EEE model uses $\beta_{4,2}(\tau)$ to represent **H9**.*

### 2.3. EEE Conceptual Framework

Having formulated nine research hypotheses regarding the environment and its effects, the EEE conceptual framework is established as shown in Figure 1. Environment is the

unique exogenous variable, while economy, scientific infrastructure, education, science and technology talent and science and technology output are endogenous variables. These exogenous and endogenous variables are all constructs and, thus, cannot be observed directly. They are so-called latent variables. For further investigation, design indicators are needed to support each construct as shown in Figure 1.

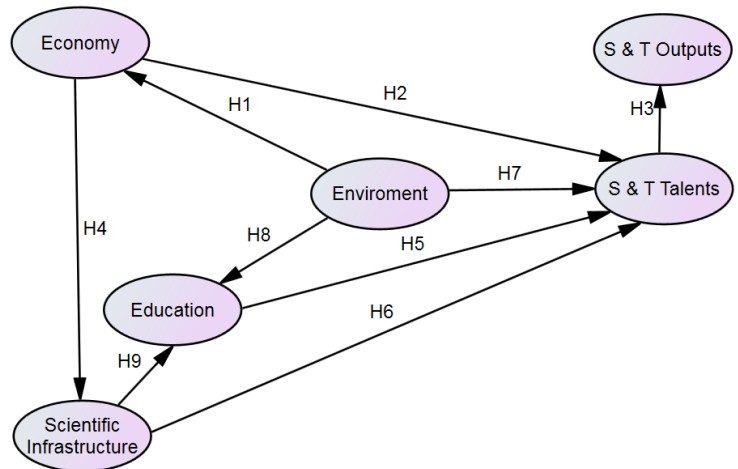

**Figure 1.** The EEE conceptual framework.

## 2.4. EEE Indicator Measurement

Each construct (latent variable) in Figure 1 is an abstract concept and needs to be measured through its corresponding indicators (observed variables), which were introduced in Section 2.1. According to the former theoretical basis research, the observed variables are listed to represent their corresponding constructs. Table 1 displays the constructs and observed variables (O.V.) of the EEE model.

**Table 1.** The constructs and observed variables of the EEE model.

| Constructs | O.V. | Original Name | Description | Unit |
|---|---|---|---|---|
| Environment (EN) | EN1 | Energy intensity | Commercial energy consumed for each dollar of GDP | kilojoules |
| | EN2 | Paper and cardboard recycling rate | Percentage of apparent consumption | % |
| | EN3 | Waste water treatment plants | Percentage of population served | % |
| | EN4 | Water consumption intensity | Water withdrawal for each 1000 US$ of GDP | m$^3$ |
| | EN5 | $CO_2$ emissions | Carbon dioxide emissions | Metric tons |
| | EN6 | Renewable energies | Share of renewables in total energy requirements | % |
| | EN7 | Total biocapacity | Biologically productive space | Hectares per capita |
| | EN8 | Sustainable development | Sustainable development is a priority in companies | Scores |
| | EN9 | Pollution problems | Pollution problems do not seriously affect the economy | Scores |
| | EN10 | Environmental laws | Environmental laws do not hinder businesses competitiveness | Scores |
| Economy (EY) | EY1 | GDP per capita | US$ per capita | US$ |
| | EY2 | Government consumption | consumption expenditure of public services provided by government departments | US$ billions |
| Education (ED) | ED1 | Education expenditure per capita | US$ per capita | US$ |
| | ED2 | English proficiency | TOEFL | Scores |
| | ED3 | Educational system | The educational system meets a competitive economy's needs | Scores |
| S and T talent (TA) | TA1 | Labor force | Employed and registered unemployed | Millions |
| | TA2 | Researchers and scientists | Researchers and scientists are attracted to the country | Scores |
| | TA3 | Brain drain | Well-educated and skilled people drain does not hinder competitiveness | Scores |
| S and T output (OU) | OU1 | Nobel prizes | Awarded since 1950 | Pieces |
| | OU2 | Patent applications | Number of applications filed by applicant's origin | Pieces |
| Scientific infrastructure (SI) | SI1 | Total expenditure on R&D | US$ millions | US$ |
| | SI2 | Scientific research legislation | Laws relating to scientific research do encourage innovation | Scores |
| | SI3 | Intellectual property rights | Intellectual property rights are adequately enforced | Scores |

The definitions of the observed variables refer to the IMD World Competitiveness Yearbook.

### 3. Data Preparation

We mainly consider two issues regarding the data used in this paper. The first issue is missing data in observed variables, which should be handled with appropriate approaches. In this paper, the median imputation approach is used, which may avoid the effects brought by extreme values of observed variables (Little et al., 1987) [29]. At length, the missing values are filled with the median of the available observations for observed variables (*OV*) containing missing data, which can be expressed as Equation (1),

$$OV_{missing}^{imputed} = median\left(OV_{missing}^{observed}\right) \tag{1}$$

Here, $OV_{missing}^{*}$ denotes these observed variables with missing data. $OV_{missing}^{observed}$ denotes the observed parts, and $OV_{missing}^{imputed}$ denotes the imputation values for the missing parts.

Having overcome the missing data problems, we standardize the data through the following Equation (2), as this is the second issue.

$$Standardized\ OV = (OV - M)/SD \tag{2}$$

Here, *M* denotes the mean value of an observed variable, and *SD* denotes the square root of an observed variable's variance.

To collect the data of 23 observed variables in Table 1, the IMD (International Institute for Management Development) World Competitiveness Yearbook was chosen for part of its data covering 61 economies. Table 2 shows the statistical characteristics of all the observed variables.

**Table 2.** The statistical characteristics of the observed variables.

| Constructs | O.V. | Min | 25% Q | Median | Mean | 75% Q | Max | Missing |
|---|---|---|---|---|---|---|---|---|
| | EN1 | 28.000 | 67.000 | 109.000 | 125.200 | 143.000 | 558.000 | 0 |
| | EN2 | 6.000 | 59.020 | 79.400 | 71.130 | 88.970 | 100.000 | 17 |
| | EN3 | 5.000 | 65.000 | 82.000 | 75.260 | 92.100 | 99.900 | 16 |
| | EN4 | 0.780 | 8.055 | 22.070 | 42.693 | 54.390 | 285.830 | 15 |
| EN | EN5 | 2.100 | 37.300 | 92.500 | 456.200 | 317.200 | 9040.700 | 0 |
| | EN6 | 0.000 | 5.500 | 12.000 | 16.020 | 22.600 | 88.300 | 0 |
| | EN7 | 0.060 | 1.190 | 2.670 | 4.129 | 4.110 | 29.550 | 2 |
| | EN8 | 3.270 | 5.270 | 5.710 | 5.867 | 6.920 | 8.340 | 0 |
| | EN9 | 3.180 | 5.040 | 6.130 | 6.096 | 7.200 | 8.790 | 0 |
| | EN10 | 3.930 | 5.350 | 5.890 | 5.990 | 6.720 | 7.710 | 0 |
| EY | EY1 | 1638.000 | 9121.000 | 19,249.000 | 27,438.000 | 42,421.000 | 102,658.000 | 0 |
| | EY2 | 1.400 | 25.100 | 56.500 | 180.700 | 140.200 | 2572.000 | 0 |
| | ED1 | 49.000 | 370.800 | 836.500 | 1442.300 | 2403.500 | 4820.000 | 1 |
| ED | ED2 | 71.000 | 84.000 | 89.000 | 88.490 | 93.000 | 100.000 | 0 |
| | ED3 | 1.880 | 3.930 | 5.450 | 5.354 | 6.700 | 8.740 | 0 |
| | TA1 | 0.190 | 2.803 | 5.600 | 20.596 | 25.168 | 157.130 | 7 |
| TA | TA2 | 0.500 | 3.234 | 4.918 | 4.711 | 5.835 | 8.978 | 0 |
| | TA3 | 1.707 | 3.667 | 4.569 | 4.719 | 5.915 | 8.269 | 0 |
| OU | OU1 | 0.000 | 0.000 | 0.000 | 9.481 | 4.750 | 285.000 | 7 |
| | OU2 | 87.000 | 729.000 | 2776.000 | 32,017.000 | 14,009.000 | 529,632.000 | 0 |
| | SI1 | 18.000 | 933.200 | 3866.000 | 25,897.800 | 15,780.800 | 502,893.000 | 3 |
| SI | SI2 | 1.756 | 3.958 | 5.177 | 5.184 | 6.565 | 8.134 | 0 |
| | SI3 | 1.171 | 4.769 | 6.172 | 6.095 | 7.609 | 9.002 | 0 |

O.V., observed variables; Min, the minimum value; 25% Q, the 25% quantile; Median, the median value; Mean, the mean value; 75% Q, the 75% quantile; Max, the maximum value; and Missing, the number of missing observations.

Furthermore, violin plots (Hintze and Nelson 1998) [30] are used to present the above 23 observed variables (as can be seen in Figure 2). As a combination of both box plots and density plots, violin plots provide two kinds of information for each observed variable. On the one hand, the length of each violin on the vertical axis provides the range of the corresponding observed values. As in a box plot, the 25th, 50th and 75th percentiles of each observed variable are calculated as the lower, median and upper hinge, and then the smallest (largest) observation greater (less) than or equal to the lower (upper) hinge $-$ (+) $1.5 * $ IQR is defined as the lower (upper) adjacent value.

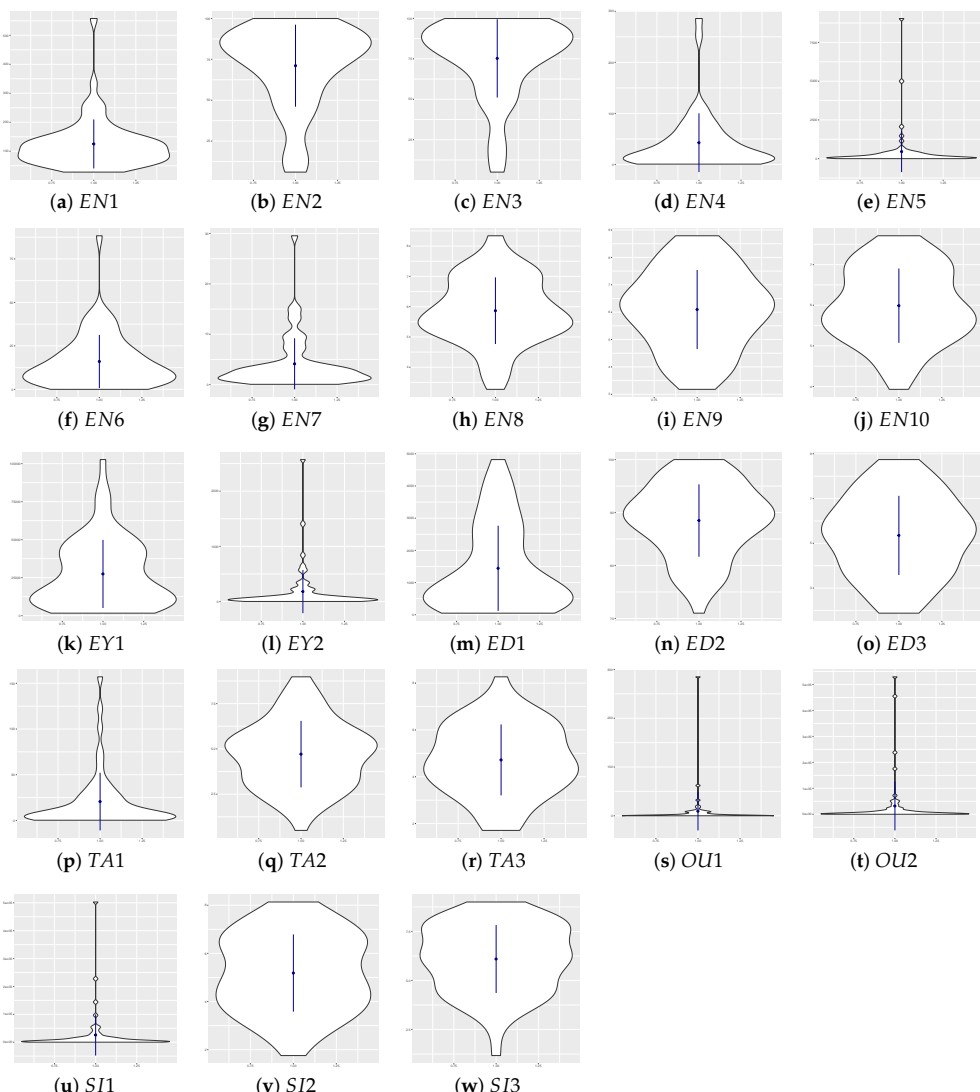

**Figure 2.** Violin plots of the observed variables.

The outliers are outside of the lower or upper adjacent values, which finally determine the length of the vertical axis of each graph. On the other hand, the shape of each violin on the horizontal axis presents the distribution of each observed variable in terms of the variability and skewness, which is similar to a density plot. It should be noted that all 23 violin plots are not required to be placed in the same planes due to the differences of measurement and scales among the different observed variables.

In Table 2, we find that $EN_2$, $EN_3$, $EN_4$, $EN_7$, $ED_1$, $TA_1$, $OU_1$ and $SI_1$ contain missing data, and their missing rates are 17/61, 16/61, 15/61, 2/61, 1/61, 7/61, 7/61 and 3/61, which equal 27.87%, 26.23%, 24.59%, 3.28%, 1.64%, 11.48%, 11.48% and 4.92%, respectively. To present the raw distribution of the data, the missing data part of each observed variable was removed before drawing each plot in Figure 2. However, concerning the quantile-type

EEE model estimation, this may lose parts of the available information for completely observed variables. Therefore, the median imputation is used to deal with missing observations before estimations. Then, the data is standardized through the data-preprocessing methods, which was mentioned before.

## 4. Methodology

### 4.1. Quantile-Type EEE Model

The establishment of this quantile-type EEE model is based on structural equation models (Bollen, 1989; Hair, et al., 2017) [31,32]. Therefore, under the framework of a classical structural equation model, the EEE model can be expressed through Equations (3) and (4).

$$\eta_{k'} = \gamma_{k'k}\xi_k + \delta_{k'} \tag{3}$$

$$X_{p_k} = \lambda_{X_{p_k}}\xi_k + \epsilon_{X_{p_k}}, Y_{p_{k'}} = \lambda_{Y_{p_{k'}}}\eta_{k'} + \epsilon_{Y_{p_{k'}}} \tag{4}$$

Here, (3) is the EEE structural model, and (4) is the EEE measurement model. Both $\xi_k$ and $\eta_{k'}$ represent the environment and its effects, such as on the economy, education and talents. The difference is, in this model, $\xi_k$ represents the $k$th exogenous latent variable: the environment. $\eta_{k'}$ represents the $k'$th endogenous latent variables, such as the economy, education and talents. $\gamma_{k'k}$ is the path coefficient linking the $k$th exogenous latent variables to the $k'$th endogenous latent variable with the error terms $\delta_{k'}$.

We assume that $\delta_{k'}$ is a random measurement error variable with mean 0 and fixed variance for the corresponding latent variable $\xi_k$. $X_{p_k}$ and $Y_{p_{k'}}$ are observed variables for the latent variables $\xi_k$ and $\eta_{k'}$, respectively. $\lambda_{X_{p_k}}$ and $\lambda_{Y_{p_{k'}}}$ represent the loading coefficients linking the observed variables to latent variables with the error terms $\epsilon_{X_{p_k}}$ and $\epsilon_{Y_{p_{k'}}}$. We assume that $\epsilon_{X_{p_k}}$ and $\epsilon_{Y_{p_{k'}}}$ are random measurement errors, which have mean 0 and are uncorrelated with their corresponding latent variables. It should be noted that, according to the distinct feature of partial least squares, none of the strict assumptions about the distributions of the random components $\delta_{k'}$, $\epsilon_{X_{p_k}}$ and $\epsilon_{Y_{p_{k'}}}$ are needed.

Based on the above EEE model, the unknown path and loading coefficients $\gamma_{k'k}$, $\lambda_{X_{p_k}}$ and $\lambda_{Y_{p_{k'}}}$ can be estimated through the most widely-used partial least squares (PLS) algorithm due to its obvious advantages, including no assumptions about the data distribution and independence. In this article, the PLS algorithm can be used as a comparison, which will be discussed in the Methodology part. Furthermore, the EEE model is developed into one kind of structural equation model in which both a path and loading coefficients can be modified into a function of the quantile level ($\tau$).

This article considers the following linear quantile regression (5) [33,34],

$$y_i = x_i^T\beta_1(\tau) + z_i^T\beta_2(\tau) + \epsilon_i, \forall \tau \in (0,1). \tag{5}$$

where $y_i$ is the response, and $x_i^T$ and $z_i^T$ are covariates. $\beta_1(\tau)$ and $\beta_2(\tau)$ are quantile-specific coefficients. $\epsilon_i$ is an independent and identically distributed error term with $P(\epsilon_i|x_i, z_i) = \tau$ for the $\tau$th quantile ($0 < \tau < 1$). We assume that the covariates ($x$ and $z$) contain the constant 1; hence, the intercept term is not written out separately.

Quantile regression estimates $\beta_\tau^* = (\beta_1(\tau), \beta_2(\tau))^T$ by solving

$$\hat{\beta}_\tau^{QR} = argmin \sum_{i=1}^{n} \varphi_\tau\left\{y_i - x_i^T\beta_1 - z_i^T\beta_2\right\}. \tag{6}$$

where $\varphi_\tau(r) = r\{\tau - I(r < 0)\}$ is the asymmetric $L_1$ loss function in quantile regression.

Thus, the quantile-type EEE model can be expressed as Equations (7) and (8). Specifically, the quantile-type EEE model consists of two parts: a quantile-type EEE structural model (7) and quantile-type EEE measurement model (8).

$$\eta_{k'} = \gamma_{k'k}(\tau)\xi_k + \delta_{k'}(\tau) \tag{7}$$

$$X_{p_k} = \lambda_{X_{p_k}}(\tau)\xi_k + \epsilon_{X_{p_k}}(\tau),\, Y_{p_{k'}} = \lambda_{Y_{p_{k'}}}(\tau)\eta_{k'} + \epsilon_{Y_{p_{k'}}}(\tau) \tag{8}$$

Here, $\xi_k$ also represents the $k$th exogenous latent variable, and $\eta_{k'}$ represents the $k'$th endogenous latent variables. Different from the classical structural equation model, the monotonicity of the right-hand-side against $\tau$ is needed. That is to say, the quantile estimation is appropriate only when the latent variables $\xi_k$ and $\eta_k$ are non-negative. $X_{p_k}$ and $Y_{p_{k'}}$ are observed variables for the latent variables $\xi_k$ and $\eta_{k'}$, respectively. $\gamma_{k'k}(\tau)$ is the path coefficient function of the quantile level ($\tau$) linking the $k$th exogenous latent variables to the $k'$th endogenous latent variable with the error terms $\delta_{k'}(\tau)$.

We assume that $\delta_{k'}(\tau)$ is a random measurement error variable with $Q\left(\delta_{k'}(\tau)\right) = 0$ and fixed variance for the corresponding latent variable $\xi_k$. The loading coefficients function of the quantile level ($\tau$) $\lambda_{X_{p_k}}(\tau)$ and $\lambda_{Y_{p_{k'}}}(\tau)$ link the observed variables to the latent variables with the error terms $\epsilon_{X_{p_k}}(\tau)$ and $\epsilon_{Y_{p_{k'}}}(\tau)$. We assume that $\epsilon_{X_{p_k}}(\tau)$ and $\epsilon_{Y_{p_{k'}}}(\tau)$ are random measurement errors with $Q\left(\epsilon_{X_{p_k}}(\tau)\right) = Q\left(\epsilon_{Y_{p_{k'}}}(\tau)\right) = 0$ and are uncorrelated with their corresponding latent variables. It should be noted that, according to the distinct feature of partial least squares, none of the strict assumptions about the distributions of random components $\delta_{k'}(\tau)$, $\epsilon_{X_{p_k}}(\tau)$ and $\epsilon_{Y_{p_{k'}}}(\tau)$ are needed.

Based on all the constructs and observed variables with the hypotheses about their relations, the establishment of this environmental effect evaluation (EEE) model is based on structural equation model theory. At length, the EEE model consists of two parts: the EEE structural model and EEE measurement model. The distinct feature of this EEE model is that both the structural model and measurement model contain quantile levels (denoted as $\tau$), which exploits quantile regression to investigate changes in the relations among constructs and between constructs and the observed variables and, thus, overcomes the classical exploration of only average effects.

Therefore, the paper proposes a quantile-type EEE model based on a quantile composite-based path-modeling approach (Dolce et al., 2021; Davino et al., 2016; Zou et al., 2008) [18,35,36]. In practice, the quantile level $\tau$ is typically chosen to be evenly spread with sufficiently dense grid points on (0, 1).

The quantile-type EEE structural model can be expressed as Equations (9)–(13),

$$EY = \beta_{1,1}(\tau)EN + \delta_1(\tau) \tag{9}$$

$$SI = \beta_{2,1}(\tau)EY + \delta_2(\tau) \tag{10}$$

$$OU = \beta_{3,1}(\tau)TA + \delta_3(\tau) \tag{11}$$

$$ED = \beta_{4,1}(\tau)EN + \beta_{4,2}(\tau)SI + \delta_4(\tau) \tag{12}$$

$$TA = \beta_{5,1}(\tau)EY + \beta_{5,2}(\tau)EN + \beta_{5,3}(\tau)ED + \beta_{5,4}(\tau)SI + \delta_5(\tau) \tag{13}$$

where the path coefficients vector $B(\tau) = (\beta_{1,1}(\tau), \beta_{2,1}(\tau), \beta_{3,1}(\tau), \beta_{4,1}(\tau), \beta_{4,2}(\tau), \beta_{5,1}(\tau), \beta_{5,2}(\tau), \beta_{5,3}(\tau), \beta_{5,4}(\tau))^T$ links different constructs. The random error term vector $\Delta = (\delta_1(\tau), \delta_2(\tau), \delta_3(\tau), \delta_4(\tau), \delta_5(\tau))^T$ has the mean vector $(0,0,0,0,0)^T$, and all the random error terms are uncorrelated with their constructs.

The quantile-type EEE measurement model, which reflects the relations among different constructs and their observed variables at quantile level $\tau$, can be expressed as Equations (14)–(19),

$$EN_j = \lambda_j^{EN}(\tau)EN + \epsilon_j^{EN}(\tau) \tag{14}$$

$$EY_i = \lambda_i^{EY}(\tau)EY + \epsilon_i^{EY}(\tau) \tag{15}$$

$$ED_l = \lambda_l^{ED}(\tau)ED + \epsilon_l^{ED}(\tau) \tag{16}$$

$$TA_m = \lambda_m^{TA}(\tau)TA + \epsilon_m^{TA}(\tau) \tag{17}$$

$$OU_v = \lambda_v^{OU}(\tau)OU + \epsilon_v^{OU}(\tau) \tag{18}$$

$$SI_k = \lambda_k^{SI}(\tau)SI + \epsilon_k^{SI}(\tau) \tag{19}$$

where $\Lambda(\tau) = \left(\lambda_j^{EN}(\tau), \lambda_i^{EY}(\tau), \lambda_l^{ED}(\tau), \lambda_m^{TA}(\tau), \lambda_v^{OU}(\tau), \lambda_k^{SI}(\tau)\right)^T$ is the loading coefficient vector linking each observed variable to its corresponding construct. Here, $i = 1, 2$, $j = 1, 2, ..., 10$, $l = 1, 2, 3$, $m = 1, 2, 3$, $v = 1, 2$ and $k = 1, 2, 3$. The random error term vector $E = \left(\epsilon_j^{EN}(\tau), \epsilon_i^{EY}(\tau), \epsilon_l^{ED}(\tau), \epsilon_m^{TA}(\tau), \epsilon_v^{OU}(\tau), \epsilon_k^{SI}(\tau)\right)^T$ has the mean vector $(0, 0, 0, 0, 0, 0)^T$, and all the random error terms are uncorrelated with their constructs.

Therefore, I propose the quantile-type EEE model based on Equations (9)–(19) and do not label the estimated coefficients ($B(\tau)$ and $\Lambda(\tau)$) in Figure 3 for brevity. It should be noted that, although the constructs cannot be directly observed and are formed by variables, the arrows still go from construct to variables, which is consistent with the well-known European customer satisfaction index (Askariazad et al., 2015) [37].

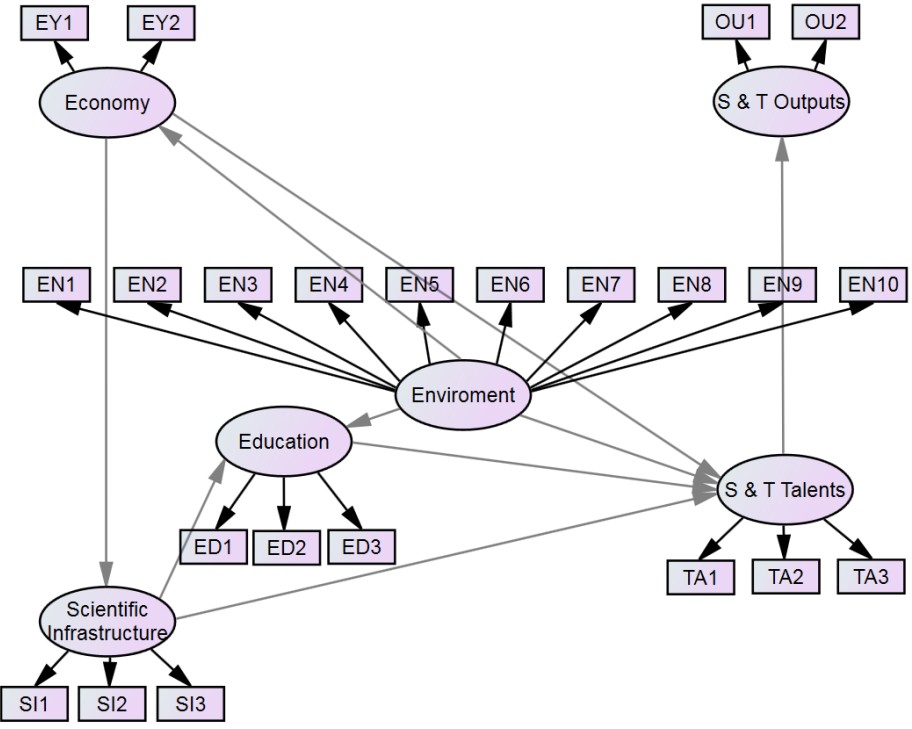

**Figure 3.** Quantile-type EEE model. EN1, energy intensity; EN2, paper and cardboard recycling rate; EN3, waste water treatment plants; EN4, water consumption intensity; EN5, $CO_2$ emissions; EN6, renewable energies; EN7, total biocapacity; EN8, sustainable development; EN9, pollution problems; EN10, environmental laws; EY1, GDP per capita; EY2, government consumption; ED1, education expenditure per capita; ED2, English proficiency; ED3, educational system; TA1, labor force; TA2, researchers and scientists; TA3, brain drain; OU1, Nobel prizes; OU2, patent applications; SI1, total expenditure on R&D; SI2, scientific research legislation; and SI3, intellectual property rights.

*4.2. Quantile-Type Path-Modeling Algorithm*

4.2.1. The Framework of the Quantile-Type Path-Modeling Algorithm

Essentially, the quantile-type path-modeling algorithm introduces quantile regression into the framework of the classical partial least squares path-modeling algorithm [38–41]. The parameter estimation of the classical partial least squares path-modeling framework follows a double approximation of the latent variables: external estimation and internal estimation, which is also the foundation of the proposed quantile-type path-modeling algorithm (which can be seen in Table 3).

**Table 3.** Double approximation procedure.

| Phrase | Description |
| --- | --- |
| External estimation | Latent variables are obtained as the product of their corresponding block of observed variables with the external weights |
| Internal estimation | Latent variables are obtained as the product of their external estimations with the internal weights |

Based on Table 3, the quantile-type path-modeling algorithm is performed as follows. To estimate the inner weights, we need to calculate the quantile correlation value between each pair of latent variables' external estimations or use a centroid scheme (that is, the sign of the quantile correlation) to obtain the inner weights. In the external weight updating procedure, the observed variables can be treated as the response variable of the latent variables. The outer weights are computed estimating the quantile regressions. The weight estimation is an iterative procedure between the external estimation and internal estimation. The whole procedure iterates between the inner and outer approximation phases until convergence of the outer vectors is achieved, i.e., until it reaches the maximum number of iterations or until the change in the outer weights of two subsequent iterations is smaller than a predefined stop criterion value at the same time.

4.2.2. Assessment Measures

Model assessment in the quantile-type EEE model is different from classical partial least squares path modeling. In other words, the standard assessment measures (Hair et al., 2019; Benitez et al., 2020) [42–44] cannot be directly applied. Based on $pseudo - R^2$ (Koenker et al., 1999), Davino et al., (2016) and Dolce et al., (2021) have proposed new quantile-type assessment measures [24,35,45].

For the inner model, $pseudo - R^2$ can be considered as an indicator of the goodness of fit to measure the explanatory power of the independent latent variables in expressing dependent latent variables. For the outer model, Dolce et al., (2021) [35] has proposed new quantile-type communality and redundancy indicators (denoted as $Com(\tau)$ and $Red(\tau)$, respectively). In the quantile-type path-modeling background, communality indicates the amount of the variance of the observed variables explained by the corresponding latent variable, which can be expressed as Equation (20). communality indicates the amount of the variance of the observed variables explained by the corresponding latent variable, which can be expressed as Equation (20).

Redundancy measures the percent of variance of the observed variables in a dependent block predicted from the corresponding explanatory latent variable, which can be written as Equations (21) and (22). Equation (21) is for each observed variable of the dependent block, while (22) is for a whole dependent block averaging the redundancies of the observed variables within the block.

$$Com(\tau)_k = \frac{1}{p_k} \sum_{p_k} pseudo - R^2(\tau) \qquad (20)$$

$$Red(\tau)_{p_k} = Com(\tau)_{p_k} pseudo - R^2(\tau)_k \tag{21}$$

$$Red(\tau)_k = \frac{1}{p_k} \sum_{p_k} Red(\tau)_{p_k} \tag{22}$$

where $p_k$ denotes the number of observed variables of the $k$th block.

## 5. Results

### 5.1. Coefficient Estimations

5.1.1. Analysis of Environmental Effects Based on the Estimated Path Coefficients

Figure 4 displays the estimated path coefficients of the quantile-type EEE model at the quantile levels 0.25, 0.50 and 0.75. It should be noted that these estimated path coefficients are also presented as the raw estimation in Table 6. This indicates that the signs of the estimated path coefficients satisfy all the theoretical hypotheses except for **H5** and **H8** at certain quantile levels. As expected, the environment has a positive impact on the economy (consistent with **H1**) and science and technology talent (consistent with **H7**), while the other variables remain unchanged at all quantile levels (0.25, 0.50 and 0.75).

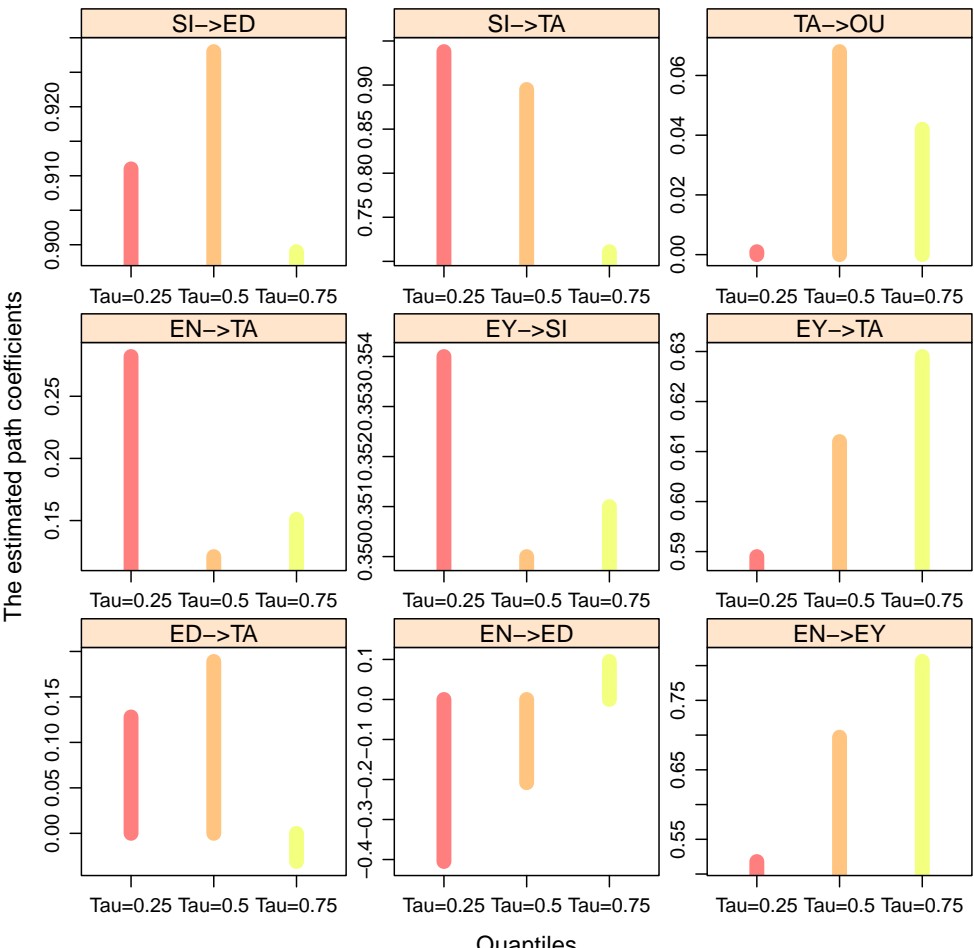

**Figure 4.** The estimated path coefficients of the quantile-type EEE model at quantile levels 0.25, 0.50 and 0.75.

The economy positively affects science and technology talent (consistent with **H2**) and scientific infrastructure (consistent with **H4**), while the environment and its other effects remain unchanged at quantile levels 0.25 to 0.75. Science and technology talent has a positive effect on science and technology output, while the environment and its other effects remain unchanged (consistent with **H3**) at all quantile levels (0.25, 0.50 and 0.75).

Scientific infrastructure positively affects science and technology talent (consistent with **H6**) and education (consistent with **H9**), while the environment and its other effects remain unchanged for quantile levels 0.25 to 0.75. At both quantile levels 0.25 and 0.50, the environment has a negative impact on education (opposed to **H8**), which negatively effects science and technology talent (opposed to **H5**) at quantile level 0.75. Their corresponding estimated path coefficients are $-0.405$, $-0.208$ and $-0.031$, respectively.

In Figure 4, in this quantile-type EEE model, no matter which quantile level is chosen, the environment (*EN*) always has a relatively large and positive impact on the economy (*EY*). The estimated path coefficients are 0.518, 0.697 and 0.806 at the quantile levels 0.25, 0.50 and 0.75, respectively. Furthermore, the impacts of the environment on the economy increase from lower quantile levels to higher quantile levels.

Compared with other relations regarding the environment and its effects, the economy (*EY*) always has a relatively large and positive impact on science and technology talent (*TA*) with the estimated path coefficients equaling 0.589, 0.612 and 0.629 at quantile levels 0.25, 0.50 and 0.75, respectively. Scientific infrastructure (*SI*) has almost the largest effects on both science and technology talent (*TA*) and education (*ED*). For scientific infrastructure (*SI*), the estimated path coefficients are 0.938, 0.895 and 0.711 for science and technology talent (*TA*) and 0.911, 0.928 and 0.899 for education (*ED*) at the quantile levels 0.25, 0.50 and 0.75, respectively.

Based on the above analysis, it can be concluded that the quantile-type path-modeling algorithm does provide different findings for the environment and its effects at different quantiles from the perspective of structural relations. As expected, the quantile-type EEE model overcomes the classical exploration of only average effects and exploits quantile regression to investigate the entire distribution of outcome variables and changes in the relations among the environment and its effects.

Another finding is that all the analyses are based on path coefficients, which are direct measures reflecting the relations between each pair of constructs. In other words, there also exist indirect effects in the quantile-type EEE structural model, such as the environment (*EN*)'s impact on science and technology output (*OU*) through science and technology talent (*TA*). According to the features of the quantile-type EEE, it can be found that the indirect effects may only exist among two kinds of relations, where the target construct is science and technology output (*OU*) or science and technology talent (*TA*). The indirect effects can be calculated by the products of all corresponding path coefficients. The indirect effects regarding the environment and its effects are presented in Table 4.

**Table 4.** The indirect effects for the environment and its effects for science and technology talent (*TA*) and science and technology output (*OU*) at quantile levels 0.25, 0.50 and 0.75.

| Path (Target: *TA*) | 0.25 | 0.50 | 0.75 | Path (Target: *OU*) | 0.25 | 0.50 | 0.75 |
|---|---|---|---|---|---|---|---|
| EN→EY→TA | 0.305 | 0.427 | 0.507 | EN→EY→TA→OU | 0.000 | 0.029 | 0.021 |
| EN→ED→TA | $-0.052$ | $-0.039$ | $-0.003$ | EN→ED→TA→OU | 0.000 | $-0.003$ | 0.000 |
| EY→SI→TA | 0.332 | 0.313 | 0.250 | EY→SI→TA→OU | 0.000 | 0.021 | 0.011 |
| SI→ED→TA | 0.117 | 0.175 | $-0.028$ | SI→ED→TA→OU | 0.000 | 0.012 | $-0.001$ |
| – | – | – | – | EN→TA→OU | 0.000 | 0.008 | 0.006 |
| – | – | – | – | EY→TA→OU | 0.001 | 0.042 | 0.026 |
| – | – | – | – | SI→TA→OU | 0.001 | 0.061 | 0.030 |
| – | – | – | – | ED→TA→OU | 0.000 | 0.013 | $-0.001$ |

Table 4 represents the following two main findings. (1) From the perspective of the environment's impacts on science and technology talent (*TA*), on the one hand, the environment (*EN*) has a relatively large and positive impact on science and technology talent (*TA*) through the economy (*EY*) at all quantile levels (0.25, 0.50 and 0.75). The indirect effects are 0.305, 0.427 and 0.507, respectively. On the other hand, the environment (*EN*) has a relatively small but negative impact on science and technology talent (*TA*) through

education ($ED$) at all quantile levels (0.25, 0.50 and 0.75). The indirect effects are $-0.052$, $-0.039$ and $-0.003$, respectively.

(2) From the perspective of the environment's impacts on science and technology output ($OU$), due to the tiny estimated path coefficients between science and technology talent ($TA$) and science and technology output ($OU$), the indirect effects are small, especially at quantile level 0.25. Specifically, the indirect effects for path $EN \rightarrow EY \rightarrow TA \rightarrow OU$ are 0.000, 0.029 and 0.021 at quantile levels 0.25, 0.50 and 0.75, respectively. The indirect effects for path $EN \rightarrow ED \rightarrow TA \rightarrow OU$ are 0.000, $-0.003$ and 0.000 at quantile levels 0.25, 0.50 and 0.75, respectively. The indirect effects for path $EN \rightarrow TA \rightarrow OU$ are 0.000, 0.008 and 0.006 at quantile levels 0.25, 0.50 and 0.75, respectively.

In this article, both direct and indirect effects are investigated based on the estimates of the quantile-type EEE model. We further present that the combination of both can be considered to cover the comprehensive effects from the environment to other constructs. For brevity, only the comprehensive effects from the environment are presented, which focuses on the main purpose of this article. For those who are interested in the effects from other constructs in this EEE model, it is easy to calculate the corresponding results, and thus they are omitted here but available upon request from the authors.

Concerning the comprehensive effects from the environment, there exist two kinds of paths: One is from the environment ($EN$) to science and technology talent ($TA$), which consists of $\{EN \rightarrow TA\}$, $\{EN \rightarrow ED \rightarrow TA\}$ and $\{EN \rightarrow EY \rightarrow TA\}$. The other path is from the environment ($EN$) to science and technology output ($OU$), which consists of $\{EN \rightarrow TA \rightarrow OU\}$, $\{EN \rightarrow ED \rightarrow TA \rightarrow OU\}$ and $\{EN \rightarrow EY \rightarrow TA \rightarrow OU\}$. Table 5 displays the comprehensive effects ($E_C$) from the environment at quantile levels 0.25, 0.50 and 0.75.

**Table 5.** The comprehensive effects ($E_C$) from the environment at quantile levels 0.25, 0.50 and 0.75.

| Source | Medium | Target | 0.25 | | | 0.5 | | | 0.75 | | |
| | | | $E_D$ | $E_I$ | $E_C$ | $E_D$ | $E_I$ | $E_C$ | $E_D$ | $E_I$ | $E_C$ |
|---|---|---|---|---|---|---|---|---|---|---|---|
| | - | | 0.282 | - | | 0.121 | - | | 0.151 | - | |
| EN | ED | TA | - | $-0.052$ | 0.535 | - | $-0.039$ | 0.509 | - | $-0.003$ | 0.655 |
| | EY | | - | 0.305 | | - | 0.427 | | - | 0.507 | |
| | TA | | - | 0.000 | | - | 0.008 | | - | 0.006 | |
| EN | ED→TA | OU | - | 0.000 | 0.000 | - | $-0.003$ | 0.034 | - | 0.000 | 0.027 |
| | EY→TA | | - | 0.000 | | - | 0.029 | | - | 0.021 | |

$E_D$, direct effects from the environment; and $E_I$, indirect effects from the environment.

### 5.1.2. Analysis of Environmental Effects Based on Estimated Loading Coefficients

In this part, the estimated loading coefficients for the quantile-type EEE measurement model are represented at the quantile levels 0.25, 0.50 and 0.75 (as seen in Figure 5). It should be noted that these estimated loading coefficients are also presented as the raw estimations in Table 7.

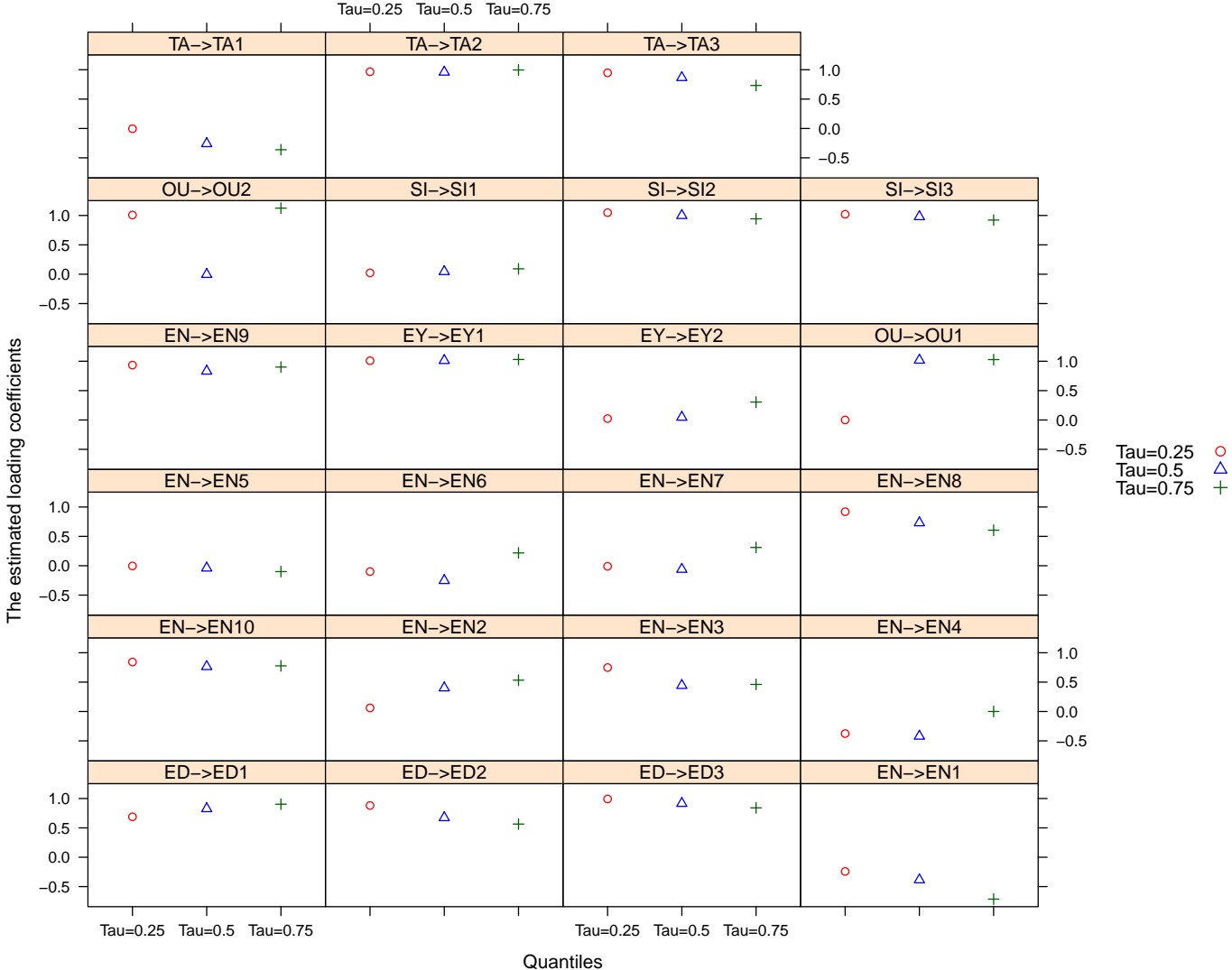

**Figure 5.** The estimated loading coefficients of the quantile-type EEE model at quantile levels 0.25, 0.50 and 0.75.

From the perspective of all quantile levels (0.25, 0.50 and 0.75), the following conclusions can be found:

(1) The estimated loading coefficients between the environment (*EN*) and pollution problems (*EN*9), environmental laws (*EN*10), sustainable development (*EN*8) and waste water treatment plants (*EN*3) are positive and relatively large, equaling {0.935, 0.842, 0.920, 0.748} at quantile level 0.25, {0.831,0.764,0.731,0.443} at quantile level 0.50 and {0.901,0.777,0.605,0.461} at quantile level 0.75, respectively. The estimated loading coefficient between the environment (*EN*) and energy intensity (*EN*1) is negative, and its absolute value is relatively large, equaling $-0.242$, $-0.385$ and $-0.713$ at quantile levels 0.25, 0.50 and 0.75, respectively.

(2) The estimated loading coefficients between scientific infrastructure (*SI*) and scientific research legislation (*SI*2) or intellectual property rights (*SI*3) are relatively large, equaling {1.048, 1.021} at quantile level 0.25, {0.999, 0.979} at quantile level 0.50 and {0.943, 0.922} at quantile level 0.75, respectively. The estimated loading coefficients between science and technology talent (*TA*) and researchers and scientists (*TA*2) or brain drain (*TA*3) are relatively large, equaling {0.966, 0.948} at quantile level 0.25, {0.959, 0.866} at quantile level 0.50 and {0.996, 0.732} at quantile level 0.75, respectively. We also find similar conclusions for the estimated loading coefficients for {EY→EY1}, {ED→ED1}, {ED→ED2} and {ED→ED3} at all quantile levels (0.25, 0.50 and 0.75).

From the perspective of each quantile level, different conclusions are found as follows:

(1) At quantile level 0.25, the estimated loading coefficient between the environment (*EN*) and water consumption intensity (*EN*4) has a relatively large absolute value but is negative, equaling −0.376. Similar conclusions can be obtained for the estimated loading coefficients for {OU→OU2}.

(2) At quantile level 0.50, the estimated loading coefficients between the environment (*EN*) and paper and cardboard recycling rate (*EN*2), science and technology output (*OU*) and Nobel prizes (*OU*1) are relatively large, equaling 0.401 and 1.018, respectively. The environment (*EN*) has relatively large but negative impacts on both renewable energies (*EN*6) and water consumption intensity (*EN*4) with estimated loading coefficients of −0.252 and −0.421, respectively. Similar conclusions can be found for the estimated loading coefficients for {TA→TA1}.

(3) At quantile level 0.75, the estimated loading coefficients between the environment (*EN*) and paper and cardboard recycling rate (*EN*2), total biocapacity (*EN*7) or renewable energies (*EN*6) are relatively large, equaling 0.533, 0.311 and 0.218, respectively. We can also find similar conclusions for the estimated loading coefficients for {OU→OU2},{OU→OU1}, {EY→EY2} and {TA→TA1}(with negative estimated loading coefficients).

### 5.2. Statistical Performance Investigations

To investigate the performance of the quantile-type path-modeling algorithm in the EEE model background, 200 bootstraps were conducted to calculate the standard errors, *p*-values and 95% confidence intervals of the estimated path and loading coefficients [46]. Table 6 displays the standard errors, *p*-values and 95% confidence intervals of the estimated path coefficients. Table 7 shows those of the estimated loading coefficients.

**Table 6.** Raw estimation (Raw), standard error (Std), P-value (P), 95 % lower confidence limit (*Low*) and 95 % upper confidence limit (*Upp*) of the estimated path coefficients at quantile levels 0.25, 0.50 and 0.75.

| $\tau$ | Path | Raw | Std | P | Low | Upp |
|---|---|---|---|---|---|---|
| | EN→EY | 0.518 | 0.133 | 0.000 | 0.252 | 0.783 |
| | EY→SI | 0.354 | 0.212 | 0.101 | −0.071 | 0.779 |
| | EN→ED | −0.405 | 0.178 | 0.027 | −0.761 | −0.049 |
| | SI→ED | 0.911 | 0.199 | 0.000 | 0.513 | 1.309 |
| 0.25 | EN→TA | 0.282 | 0.113 | 0.016 | 0.055 | 0.509 |
| | EY→TA | 0.589 | 0.202 | 0.005 | 0.184 | 0.994 |
| | SI→TA | 0.938 | 0.198 | 0.000 | 0.543 | 1.334 |
| | ED→TA | 0.128 | 0.160 | 0.427 | −0.193 | 0.449 |
| | TA→OU | 0.001 | 0.004 | 0.866 | −0.007 | 0.009 |
| | EN→EY | 0.697 | 0.067 | 0.000 | 0.564 | 0.830 |
| | EY→SI | 0.350 | 0.165 | 0.038 | 0.020 | 0.680 |
| | EN→ED | −0.208 | 0.179 | 0.252 | −0.567 | 0.152 |
| | SI→ED | 0.928 | 0.154 | 0.000 | 0.620 | 1.236 |
| 0.50 | EN→TA | 0.121 | 0.154 | 0.434 | −0.187 | 0.429 |
| | EY→TA | 0.612 | 0.172 | 0.001 | 0.269 | 0.956 |
| | SI→TA | 0.895 | 0.171 | 0.000 | 0.552 | 1.238 |
| | ED→TA | 0.189 | 0.186 | 0.316 | −0.185 | 0.562 |
| | TA→OU | 0.068 | 0.022 | 0.003 | 0.024 | 0.112 |
| | EN→EY | 0.806 | 0.094 | 0.000 | 0.618 | 0.993 |
| | EY→SI | 0.351 | 0.142 | 0.016 | 0.067 | 0.636 |
| | EN→ED | 0.095 | 0.195 | 0.629 | −0.296 | 0.486 |
| | SI→ED | 0.899 | 0.150 | 0.000 | 0.599 | 1.199 |
| 0.75 | EN→TA | 0.151 | 0.161 | 0.353 | −0.172 | 0.474 |
| | EY→TA | 0.629 | 0.177 | 0.001 | 0.274 | 0.984 |
| | SI→TA | 0.711 | 0.235 | 0.004 | 0.241 | 1.181 |
| | ED→TA | −0.031 | 0.277 | 0.912 | −0.586 | 0.525 |
| | TA→OU | 0.042 | 0.107 | 0.694 | −0.172 | 0.256 |

**Table 7.** Raw estimation (Raw), standard error (Std), P-value (P), 95 % lower confidence limit (*Low*) and 95 % upper confidence limit (*Upp*) of the estimated loading coefficients at quantile levels 0.25, 0.50 and 0.75.

| | 0.25 | | | | | 0.5 | | | | | 0.75 | | | | |
|---|---|---|---|---|---|---|---|---|---|---|---|---|---|---|---|
| | Raw | Std | P | Low | Upp | Raw | Std | P | Low | Upp | Raw | Std | P | Low | Upp |
| EN→EN1 | −0.242 | 0.078 | 0.003 | −0.399 | −0.086 | −0.385 | 0.124 | 0.003 | −0.633 | −0.137 | −0.713 | 0.184 | 0.000 | −1.081 | −0.345 |
| EN→EN2 | 0.061 | 0.253 | 0.811 | −0.445 | 0.567 | 0.401 | 0.088 | 0.000 | 0.226 | 0.576 | 0.533 | 0.127 | 0.000 | 0.279 | 0.786 |
| EN→EN3 | 0.748 | 0.235 | 0.002 | 0.279 | 1.218 | 0.443 | 0.140 | 0.003 | 0.162 | 0.723 | 0.461 | 0.120 | 0.000 | 0.221 | 0.700 |
| EN→EN4 | −0.376 | 0.106 | 0.001 | −0.588 | −0.165 | −0.421 | 0.056 | 0.000 | −0.534 | −0.308 | 0.000 | 0.155 | 1.000 | −0.309 | 0.309 |
| EN→EN5 | −0.004 | 0.007 | 0.546 | −0.018 | 0.009 | −0.040 | 0.025 | 0.114 | −0.090 | 0.010 | −0.100 | 0.056 | 0.079 | −0.211 | 0.012 |
| EN→EN6 | −0.100 | 0.115 | 0.391 | −0.330 | 0.131 | −0.252 | 0.183 | 0.175 | −0.619 | 0.116 | 0.218 | 0.222 | 0.330 | −0.226 | 0.661 |
| EN→EN7 | −0.010 | 0.051 | 0.853 | −0.112 | 0.093 | −0.061 | 0.115 | 0.600 | −0.290 | 0.169 | 0.311 | 0.231 | 0.184 | −0.152 | 0.774 |
| EN→EN8 | 0.920 | 0.134 | 0.000 | 0.651 | 1.188 | 0.731 | 0.132 | 0.000 | 0.467 | 0.995 | 0.605 | 0.102 | 0.000 | 0.400 | 0.809 |
| EN→EN9 | 0.935 | 0.079 | 0.000 | 0.776 | 1.093 | 0.831 | 0.066 | 0.000 | 0.699 | 0.964 | 0.901 | 0.066 | 0.000 | 0.770 | 1.033 |
| EN→EN10 | 0.842 | 0.102 | 0.000 | 0.638 | 1.047 | 0.764 | 0.129 | 0.000 | 0.507 | 1.021 | 0.777 | 0.086 | 0.000 | 0.605 | 0.950 |
| EY→EY1 | 1.009 | 0.009 | 0.000 | 0.991 | 1.026 | 1.012 | 0.002 | 0.000 | 1.010 | 1.015 | 1.030 | 0.003 | 0.000 | 1.025 | 1.035 |
| EY→EY2 | 0.024 | 0.021 | 0.264 | −0.019 | 0.066 | 0.046 | 0.048 | 0.347 | −0.051 | 0.142 | 0.305 | 0.252 | 0.230 | −0.198 | 0.809 |
| SI→SI1 | 0.023 | 0.009 | 0.012 | 0.005 | 0.040 | 0.044 | 0.023 | 0.058 | −0.002 | 0.089 | 0.092 | 0.114 | 0.422 | −0.136 | 0.320 |
| SI→SI2 | 1.048 | 0.036 | 0.000 | 0.976 | 1.120 | 0.999 | 0.039 | 0.000 | 0.921 | 1.076 | 0.943 | 0.037 | 0.000 | 0.869 | 1.016 |
| SI→SI3 | 1.021 | 0.039 | 0.000 | 0.944 | 1.098 | 0.979 | 0.046 | 0.000 | 0.886 | 1.072 | 0.922 | 0.040 | 0.000 | 0.842 | 1.001 |
| ED→ED1 | 0.688 | 0.110 | 0.000 | 0.468 | 0.908 | 0.826 | 0.091 | 0.000 | 0.643 | 1.008 | 0.903 | 0.089 | 0.000 | 0.726 | 1.080 |
| ED→ED2 | 0.881 | 0.113 | 0.000 | 0.654 | 1.107 | 0.673 | 0.120 | 0.000 | 0.433 | 0.914 | 0.565 | 0.084 | 0.000 | 0.396 | 0.734 |
| ED→ED3 | 0.992 | 0.082 | 0.000 | 0.828 | 1.156 | 0.916 | 0.077 | 0.000 | 0.762 | 1.069 | 0.841 | 0.068 | 0.000 | 0.705 | 0.977 |
| TA→TA1 | −0.004 | 0.029 | 0.883 | −0.062 | 0.054 | −0.259 | 0.095 | 0.008 | −0.448 | −0.070 | −0.362 | 0.159 | 0.026 | −0.679 | −0.044 |
| TA→TA2 | 0.966 | 0.084 | 0.000 | 0.798 | 1.133 | 0.959 | 0.053 | 0.000 | 0.854 | 1.065 | 0.996 | 0.064 | 0.000 | 0.868 | 1.125 |
| TA→TA3 | 0.948 | 0.061 | 0.000 | 0.827 | 1.069 | 0.866 | 0.053 | 0.000 | 0.761 | 0.972 | 0.732 | 0.105 | 0.000 | 0.522 | 0.942 |
| OU→OU1 | 0.000 | 0.055 | 1.000 | −0.110 | 0.110 | 1.018 | 0.064 | 0.000 | 0.891 | 1.146 | 1.028 | 0.028 | 0.000 | 0.972 | 1.084 |
| OU→OU2 | 1.008 | 0.000 | 0.000 | 1.008 | 1.008 | −0.004 | 0.541 | 0.995 | −1.086 | 1.079 | 1.125 | 1.374 | 0.416 | −1.624 | 3.874 |

Table 6 consists of three layers in total, which display the standard errors, *p*-value, 95% lower confidence limit and 95 % upper confidence limit of the corresponding estimates in the path coefficients, respectively. Each layer represents those corresponding results at each quantile level.

As expected, at different quantile levels, we find different conclusions regarding the standard errors, *p*-values and 95% confidence intervals. From the perspective of the standard errors, almost all the path coefficient estimators have relatively small standard errors (less than 0.200) except for $EY \rightarrow SI$, $EY \rightarrow TA$ and $SI \rightarrow TA$ at quantile level 0.25 and $SI \rightarrow TA$ and $ED \rightarrow TA$ at quantile level 0.75. From the perspective of the P-values, the estimated path coefficients for $EN \rightarrow EY$, $SI \rightarrow ED$, $EY \rightarrow TA$ and $SI \rightarrow TA$ have obviously small *p*-values compared with 0.050 at all quantile levels (0.25, 0.50 and 0.75).

In addition, the estimated path coefficients for $EN \rightarrow ED$ and $EN \rightarrow TA$ have obviously small *p*-values compared with 0.050 at quantile level 0.25, and so do $TA \rightarrow OU$ at quantile level 0.50 and $EY \rightarrow SI$ at quantile level 0.75. It should be noted that, although the *p*-values for other estimated path coefficient estimators are relatively larger than 0.05, the relations among the corresponding constructs in the quantile-type EEE structural model are kept due to their importance in supporting environmental effect evaluation in the real world.

In Table 7, most standard errors of the estimated loading coefficients are relatively small at quantile levels 0.25, 0.50 and 0.75 except for $\{EN \rightarrow EN2\}$ and $\{EN \rightarrow EN3\}$ at quantile level 0.25, $\{OU \rightarrow OU2\}$ at quantile level 0.50 and $\{EN \rightarrow EN6\}$, $\{EN \rightarrow EN7\}$, $\{EY \rightarrow EY2\}$ and $\{OU \rightarrow OU2\}$ at quantile level 0.75. These standard errors are relatively large (more than 0.200). The *p*-values for $\{EN \rightarrow EN5\}$, $\{EN \rightarrow EN6\}$, $\{EN \rightarrow EN7\}$ and $\{EY \rightarrow EY2\}$ are larger than 0.050 at all quantile levels (0.25, 0.50 and 0.75).

At quantile level 0.25, the *p*-values for $\{EN \rightarrow EN2\}$, $\{OU \rightarrow OU1\}$ and $\{TA \rightarrow TA1\}$ are larger than 0.050. At quantile level 0.50, the *p*-value for $\{OU \rightarrow OU2\}$ is larger than 0.050. At quantile level 0.75, the *p*-values for $\{EN \rightarrow EN4\}$, $\{OU \rightarrow OU2\}$ and $\{SI \rightarrow SI1\}$ are larger than 0.050. It should be noted that, although the *p*-values for these estimated loading coefficient estimators are relatively larger than 0.05, we still maintain the relations among the constructs and their corresponding observed variables in the quantile-type EEE measurement model due to their importance in supporting environmental effect evaluation in the real world.

*5.3. Model Assessment and Validation*

This part investigates the assessment of the internal and external estimation parts of the the quantile-type EEE model. To accomplish the above analysis concerns in the quantile background, a series of assessment assignments were performed separately at each quantile level. Regarding the measurement model, the amount of variability of the endogenous constructs explained by their explanatory constructs is measured by the $Pseudo.R^2$ results.

A synthesis of the evaluations regarding the whole inner model can be obtained by the average of all the $Pseudo.R^2$ results. The evaluation of endogenous blocks also covers the external part of the model through the *Redundancy* measures, expressing the percentage of the variance of the observed variables in the endogenous blocks, as predicted from explanatory constructs. The averages of the observed variable redundancies of the endogenous blocks can be denoted as $Redundancy_{Block}$.

The communality values for each block, related to each observed variable and to the whole block, are stored in *Communality* and $Communality_{Block}$, respectively. We present the $Pseudo.R^2$ of the quantile-type EEE model at the quantile levels 0.25, 0.50 and 0.75 in Table 8 and the *Redundancy* and *Communality* of the quantile-type EEE model at quantile levels 0.25, 0.50 and 0.75 in Tables 9 and 10, respectively.

Table 8 represents the $Pseudo.R^2$ for each endogenous construct at each quantile level. Clearly, at all quantile levels, only *OU* has very tiny $Pseudo.R^2$ values, which illustrates that its unique explanatory construct *TA* provides very limited contributions to the explanation of *OU*. Compared with other endogenous constructs, it can be understood that *OU* is only directly affected by *TA*. The limited number of selected factors may lead to the tiny $Pseudo.R^2$ values. From the perspective of each quantile of interest, the explanatory power of its exogenous variables is greater in the group of countries with middle- and high-level economic (*EY*) conditions.

The same conclusions can be found for *SI*, *ED*, *TA* and *OU*. For the synthesis of the evaluations regarding the whole inner model, those who are interested can calculate the average of all the $Pseudo.R^2$ values at each quantile level, which can be seen in the last column of Table 8.

**Table 8.** $Pseudo.R^2$ of the quantile-type EEE model at quantile levels 0.25, 0.50 and 0.75.

|  |  | EY | SI | ED | TA | OU | Average of $Pseudo.R^2$ |
|---|---|---|---|---|---|---|---|
|  | 0.25 | 0.314 | 0.295 | 0.501 | 0.556 | 0.000 | 0.333 |
| $Pseudo.R^2$ | 0.50 | 0.433 | 0.387 | 0.572 | 0.603 | 0.041 | 0.407 |
|  | 0.75 | 0.403 | 0.372 | 0.636 | 0.604 | 0.002 | 0.403 |

**Table 9.** *Redundancy* of the quantile-type EEE model at quantile levels 0.25, 0.50 and 0.75.

| SI | 0.25 | 0.50 | 0.75 | TA | 0.25 | 0.50 | 0.75 | ED | 0.25 | 0.50 | 0.75 |
|---|---|---|---|---|---|---|---|---|---|---|---|
| SI1 | 0.004 | 0.009 | 0.002 | TA1 | 0.000 | 0.045 | 0.048 | ED1 | 0.184 | 0.303 | 0.380 |
| SI2 | 0.235 | 0.313 | 0.297 | TA2 | 0.328 | 0.387 | 0.374 | ED2 | 0.186 | 0.177 | 0.215 |
| SI3 | 0.234 | 0.301 | 0.286 | TA3 | 0.345 | 0.315 | 0.284 | ED3 | 0.290 | 0.321 | 0.356 |
| $SI_{Block}$ | 0.158 | 0.208 | 0.195 | $TA_{Block}$ | 0.225 | 0.249 | 0.235 | $ED_{Block}$ | 0.220 | 0.267 | 0.317 |
| EY | 0.25 | 0.50 | 0.75 | OU | 0.25 | 0.50 | 0.75 | - | - | - | - |
| EY1 | 0.302 | 0.425 | 0.387 | OU1 | 0.000 | 0.034 | 0.002 | - | - | - | - |
| EY2 | 0.002 | 0.004 | 0.014 | OU2 | 0.000 | 0.000 | 0.000 | - | - | - | - |
| $EY_{Block}$ | 0.152 | 0.215 | 0.200 | $OU_{Block}$ | 0.000 | 0.017 | 0.001 | - | - | - | - |

$SI_{Block}$ and $Redundancy_{Block}$ for SI and its block of observed variables; $TA_{Block}$ and $Redundancy_{Block}$ for TA and its block of observed variables; $ED_{Block}$ and $Redundancy_{Block}$ for ED and its block of observed variables; $EY_{Block}$ and $Redundancy_{Block}$ for EY and its block of observed variables; and $OU_{Block}$ and $Redundancy_{Block}$ for OU and its block of observed variables.

**Table 10.** *Communality* of the quantile-type EEE model at quantile levels 0.25, 0.50 and 0.75.

|  | 0.25 | 0.50 | 0.75 |  | 0.25 | 0.50 | 0.75 |
|---|---|---|---|---|---|---|---|
| EN1 | 0.163 | 0.167 | 0.206 | EY1 | 0.964 | 0.981 | 0.961 |
| EN2 | 0.005 | 0.187 | 0.162 | EY2 | 0.006 | 0.010 | 0.033 |
| EN3 | 0.216 | 0.205 | 0.130 | $EY_{Block}$ | 0.485 | 0.496 | 0.497 |
| EN4 | 0.158 | 0.208 | 0.000 | ED1 | 0.368 | 0.530 | 0.598 |
| EN5 | 0.003 | 0.013 | 0.034 | ED2 | 0.372 | 0.309 | 0.338 |
| EN6 | 0.019 | 0.018 | 0.021 | ED3 | 0.579 | 0.561 | 0.559 |
| EN7 | 0.001 | 0.001 | 0.069 | $ED_{Block}$ | 0.440 | 0.467 | 0.498 |
| EN8 | 0.326 | 0.284 | 0.318 | TA1 | 0.000 | 0.075 | 0.079 |
| EN9 | 0.573 | 0.592 | 0.673 | TA2 | 0.591 | 0.641 | 0.619 |
| EN10 | 0.463 | 0.412 | 0.446 | TA3 | 0.621 | 0.523 | 0.471 |
| $EN_{Block}$ | 0.193 | 0.209 | 0.206 | $TA_{Block}$ | 0.404 | 0.413 | 0.390 |
| SI1 | 0.015 | 0.024 | 0.006 | OU1 | 0.000 | 0.820 | 0.903 |
| SI2 | 0.795 | 0.809 | 0.799 | OU2 | 1.000 | 0.001 | 0.100 |
| SI3 | 0.795 | 0.779 | 0.770 | $OU_{Block}$ | 0.500 | 0.411 | 0.501 |
| $SI_{Block}$ | 0.535 | 0.537 | 0.525 | - | - | - | - |

$EN_{Block}$ and *Communality*$_{Block}$ for EN and its block of observed variables; $SI_{Block}$ and *Communality*$_{Block}$ for SI and its block of observed variables; $TA_{Block}$ and *Communality*$_{Block}$ for TA and its block of observed variables; $ED_{Block}$ and *Communality*$_{Block}$ for ED and its block of observed variables; $EY_{Block}$ and *Communality*$_{Block}$ for EY and its block of observed variables; and $OU_{Block}$ and *Communality*$_{Block}$ for OU and its block of observed variables.

In Table 9, we find that, from the perspective of the external part of the quantile-type EEE model, different *Redundancy* values measure different percents of the variance of the observed variables in their corresponding endogenous blocks that are predicted from the explanatory constructs related to the endogenous constructs SI, TA, ED, EY and OU. From the perspective of each quantile of interest, the *Redundancy* is greater in the group of countries with middle-quantile-level scientific infrastructure ($SI$) or science and technology output ($OU$) conditions.

The *Redundancy* is greater in the group of countries with the highest education ($ED$) conditions. From the perspective of *Redundancy*$_{Block}$, the *Redundancy* is greater in the group of countries with middle-quantile-level scientific infrastructure ($SI$) or science and technology output ($OU$) conditions. *Redundancy*$_{Block}$ is also greater in the group of countries with the highest education ($ED$) conditions, while, for other constructs, it is greater in the group of countries with middle-quantile-level conditions.

Table 10 displays that, at all quantile levels (0.25, 0.50 and 0.75), the variance of $EN_9$ is explained most by its corresponding construct $EN$ when compared with the other observed variables. The variance of $SI_2$ is explained most by its corresponding construct $SI$ when compared with the other observed variables. The variance of $EY_1$ is explained most by its corresponding construct $EY$ when compared with the other observed variable $EY_2$.

For $ED$, $TA$ and $OU$, the conclusions are different at each quantile level. The variance of $ED_3$ is explained most by its corresponding construct $ED$ when compared with the other observed variables at quantile levels 0.25 and 0.50, while the variance of $ED_1$ is explained most by its corresponding construct $ED$ when compared with the other observed variables at quantile level 0.75. The variance of $TA_2$ is explained most by its corresponding construct $TA$ when compared with other the observed variables at quantile levels 0.50 and 0.75, while the variance of $TA_3$ is explained most by its corresponding construct $TA$ when compared with the other observed variables at quantile level 0.25.

The variance of $OU_1$ is explained most by its corresponding construct $OU$ when compared with the other observed variables at quantile levels 0.50 and 0.75, while the variance of $OU_2$ is explained most by its corresponding construct $OU$ when compared with the other observed variable $OU_1$ at quantile level 0.25. From the perspective of *Communality*$_{Block}$, which can be calculated for each block using the average of the observed variable *Communality*, there do not exist obvious differences in the block of each construct at the different quantile levels.

## 6. Discussions

This article focused on investigating environmental effect evaluation from the perspective of qualitative analysis using a quantile-type path model and algorithm. The environment affects many things, including the economy, education, science and technology talent, scientific infrastructure and science and technology output. Therefore, I proposed the quantile-type EEE model and applied a quantile-type path-modeling algorithm under the premise of nine theoretical research hypotheses.

The investigations showed that the environment had indispensable direct impacts on the economy, education and science and technology talent and had indirect effects on scientific infrastructure and science and technology output. Due to the advantages of quantile regression, the quantile-type EEE model and path-modeling algorithm overcame the classical exploration of only average effects and instead captured different relations regarding the environment and its effects at the quantiles of interest.

In the future, additional investigations will be conducted to further clarify these findings, and more constructs will be introduced into the model to deeply investigate the effects of the environment. In this case, the investigation on only the environmental effects may not be sufficient, and environmental factors will be considered at the same time. When both the environmental effects and factors become dynamic over time and when the coefficients of the EEE model vary, more promising statistical models and algorithms are greatly needed (Fan et al., 2008; Chiou et al., 2012; Cheng et al., 2022; Wei et al., 2022) [47–50].

Another potential direction is establishing a comprehensive indicator based on a new kind of environmental-effects model using a hierarchical latent-variable model. Specifically, environmental effects will be evaluated through certain sub-effects or dimensions. Furthermore, other kinds of path-modeling algorithms considering more complex model estimations and applications should be investigated in near future [51–54].

**Funding:** This work was supported by grants from the Natural Science Foundation of China (72001197), National Statistical Science Research Project of National Bureau of Statistics (2021LY052) and the Fundamental Research Funds for the Central Universities, and the Research Funds of Renmin University of China (16XNH102).

**Institutional Review Board Statement:** This article does not contain any studies with human participants or animals performed by any of the authors.

**Informed Consent Statement:** Not applicable.

**Data Availability Statement:** All the data analyzed during this study are included in this article. The datasets analyzed during the current study are available in the International Institute for Management Development (IMD) World Competitiveness Yearbook, https://worldcompetitiveness.imd.org/, accessed on 3 February 2023. I used R software for programming.

**Acknowledgments:** The author is very grateful to the anonymous reviewers for their insightful comments and to the interviewees for participating in this study. The author's work was supported by the National Natural Science Foundation of China (72001197), National Statistical Science Research Project of National Bureau of Statistics (2021LY052) and the Fundamental Research Funds for the Central Universities, and the Research Funds of Renmin University of China (16XNH102). The author wants to thank his parents, his wife Yujie Liu and his cute babies: QQ and QD.

**Conflicts of Interest:** The author declares no competing interests. The author declares that he has no known competing financial interests or personal relationships that could have appeared to influence the work reported in this article.

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
