# Peer review of "Environmental Effect Evaluation: A Quantile-Type Path-Modeling Approach"

_sustainability, doi:10.3390/su15054399_

Round 1
Reviewer 1 Report
The article is good as an experimentation with a new methdology. However few clarifications and improvements will strengthen the paper.
1. Clarification: why is government consumption taken for explaining country's economic status? Government consumption can also be through deficit, which will not reflect the true economic status of the country.
2. Clarification: Education variables talk of English proficiency as one of the variables. Why was enrollment in higher education not taken as one of the variables of education?
3. Clarification: Hypotheses 7 and 8 are unclear. The assumptions made for these two hypotheses require explanation. For example, how can environment development parameters impact English Proficiency? The relationship emerges significant in the model.
4. The mathematics is extensive. The explanation of the model is adequate.
5. Improvement: The language can be improved with the reduction of passive voice words like 'it' 'it is' 'that' that is' 'this' 'there' 'there is' 'these' and so forth. Extensive use of 'we' is seen as well, and can be modified with an active voice sentence construction.
6: Improvement: The results require better explanation as some results look spurious. Author(s) can take literature support to discuss the results.
Author Response
Dear reviewers and editors,
Thank you for your patience and kindness. The following are point-to-point responses respectively:
Comments and Suggestions for Authors
The article is good as an experimentation with a new methdology. However few clarifications and improvements will strengthen the paper.
- Clarification: why is government consumption taken for explaining country's economic status? Government consumption can also be through deficit, which will not reflect the true economic status of the country.
Response: Thank you for pointing all these out. I have explained the reason in section 2.1 and highlighted in blue color in my revised manuscript as follows.
Government consumption refers to the consumption expenditure of public services provided by government departments for the whole society and the net expenditure of goods and services provided to households free of charge or at a lower price. Many experts investigate the relationship between government consumption and economic growth and find the tight association between them (Wu et al., 2009).
- Clarification: Education variables talk of English proficiency as one of the variables. Why was enrollment in higher education not taken as one of the variables of education?
Response: Thank you for pointing all these out. I have explained the reason in section 2.1 and highlighted in blue color in my revised manuscript as follows.
To investigate the environment effects based on 61 economies, English proficiency must be an indicator reflecting education internationalization level. Here we do not consider enrollment in higher education mainly due to the following reason: The overall plan for deepening the reform of education evaluation in the new era requires that Party committees and governments at all levels shall not use the enrollment index or the enrollment rate of the college entrance examination as the assessment standard, and it is strictly prohibited to publish, publicize and hype the ”top scorer” and enrollment rate of the college entrance examination. In our paper, we do not use enrollment in higher education, and consider education expenditure per capita, English proficiency, and educational system to explain the definition of education.
- Clarification: Hypotheses 7 and 8 are unclear. The assumptions made for these two hypotheses require explanation. For example, how can environment development parameters impact English Proficiency? The relationship emerges significant in the model.
Response: Thank you for pointing all these out. Firstly, we have revised the environment development into environment, which provides surroundings or natural conditions to improve the education status. Secondly, English Proficiency is an observed indicator supporting education, environment parameters actually impact education including English Proficiency levels.
- The mathematics is extensive. The explanation of the model is adequate.
Response: Thank you for pointing all these out. I have reorganized the methodology part and explain the mathematical model more specifically, which has been highlighted in blue color.
- Improvement: The language can be improved with the reduction of passive voice words like 'it' 'it is' 'that' that is' 'this' 'there' 'there is' 'these' and so forth. Extensive use of 'we' is seen as well, and can be modified with an active voice sentence construction.
Response: Thank you for pointing all these out. I have revised the corresponding parts especially ‘we’, ‘these’ and so on. There are so many places that I revised, thus I only highlight part of the revised parts in blue color.
6: Improvement: The results require better explanation as some results look spurious. Author(s) can take literature support to discuss the results.
Response: Thank you for pointing all these out. I have tried to added some literatures on related topic such as Wu L. L. and Yin Y. (2009). Coordination between investment and consumption and economic growth: An empirical study from China's regional panel data. Research on Financial and Economic Issues, 5, 12-17. And in discussions part, I also added more literature investigation works will be carried out to further clarify our findings, which has been highlighted in blue color.

Reviewer 2 Report
The paper discusses a structural equation model for the environment and its effects on economy, education, etc. The quantile estimation is considered for the prediction of the tail behavior of the response variable. Below are some comments:
1. Need to polish the English. For example, in the abstract, line 6, "our article proofs the hypothesis ..."
2. In Equation (1) and (2), the notation "*" seems to represent the sum of product over the index k. Though I can guess the meaning, it is a little bit confusing at first glance.
3. In order that Equation (3) and (4) give suitable quantile estimation, some conditions on the monotonicity of the right-hand-side against tau seems needed. That is to say, the quantile estimation is appropriate only when the latent variables xi_k and eta_k are non-negative. Please comment on this.
4. In p.6, the hypotheses should be clearly stated in terms of the model parameters.
5. Bad English: in p.8 the first sentence of section 3.1. "two data problems" means "two issues about the data used in our paper"? You described the "first data problem (first issue)". Then, what is the second one?
6. Need to improve the logic flow and organization of the paper. For example, some technical parts in the introduction can be moved to the theory part. Data preparation should be mentioned before discussing the theory.
Author Response
Dear reviewers and editors,
Thank you for your patience and kindness. The following are point-to-point responses respectively:
Comments and Suggestions for Authors
The paper discusses a structural equation model for the environment and its effects on economy, education, etc. The quantile estimation is considered for the prediction of the tail behavior of the response variable. Below are some comments:
- Need to polish the English. For example, in the abstract, line 6, "our article proofs the hypothesis ..."
Response: Thank you for pointing all these out. I have revised the corresponding parts and highlighted in blue color. Please receive my revised manuscript.
- In Equation (1) and (2), the notation "*" seems to represent the sum of product over the index k. Though I can guess the meaning, it is a little bit confusing at first glance.
Response: Thank you for pointing all these out. I have revised the corresponding parts and highlighted in blue color. Please receive my revised manuscript.
- In order that Equation (3) and (4) give suitable quantile estimation, some conditions on the monotonicity of the right-hand-side against tau seems needed. That is to say, the quantile estimation is appropriate only when the latent variables xi_k and eta_k are non-negative. Please comment on this.
Response: Thank you for pointing all these out. I have added the corresponding parts and highlighted in blue color as follows.
Different from classical structural equation model, the monotonicity of the right-hand-side against τ is needed. That is to say, the quantile estimation is appropriate only when the latent variables ξk and ηk are non-negatively.
- In p.6, the hypotheses should be clearly stated in terms of the model parameters.
Response: Thank you for pointing all these out. I have added the corresponding parts following each hypothesis and highlighted in blue color, which helps to show the corresponding relations between model parameters and hypotheses.
- Bad English: in p.8 the first sentence of section 3.1. "two data problems" means "two issues about the data used in our paper"? You described the "first data problem (first issue)". Then, what is the second one?
Response: Thank you for pointing all these out. I have revised the corresponding parts and highlighted in blue color as follows.
We mainly consider two issues about the data used in our paper. The first issue is missing data in observed variables, which should be handled with appropriate approaches…Having accomplished missing data problems, we standardize our data through the following equation (2), which is the second issue.
- Need to improve the logic flow and organization of the paper. For example, some technical parts in the introduction can be moved to the theory part. Data preparation should be mentioned before discussing the theory.
Response: Thank you for pointing all these out. I have totally revised the logic flow and organization of the paper, which has been highlighted in blue color.

Reviewer 3 Report
– First, you should precisely define what do you mean by “Environment development”. What kind of environment. We can have natural, social, economic, legal … etc. environment. Most probably you have a natural environment in mind. But what is its “development”? If there is the development, it should be some “goals” of this development. What are they? Maybe “natural environment state (or status, or condition)” would be better.
Page 3 – In my opinion, a letter x is not a good choice for variables. In many researches it is used as a symbol for random component. For your assumptions about random component you should add symmetry of its distribution (if you don’t want to have normality).
Page 5 – There is something missing in the sentence starting with “Different from economy, …”
Page 5 – The IMD list of indicators shows that we are talking about “environmental protection” not “development”. How do you measure “sustainable development” (kind of magic key phrase) and “environmental laws”?
Page 7 – Hypotheses have not been “identified”. They were “formulated”
Page 8, Table 1 – You should add “measuremet unit” column. Some of the variables are dummies (I suppose). How they were coded? What is “Government consumption”? When you use data from 61 economies – it should be expressed per capita (or somehow made comparable) and it should be stated in description or measurement column.
Page 11, Figure 3 – The figure is not clear. Everything should be explained, eg. ED1, ED2, ED3 – even just by refering to Table 1. Why arrows go from construct to variables when – if fact – the values constructs are “formed” by variables.
You should provide more arguments for usage of quantile regression, not just “basic” regression model.
Author Response
Dear reviewers and editors,
Thank you for your patience and kindness. The following are point-to-point responses respectively:
Comments and Suggestions for Authors
– First, you should precisely define what do you mean by “Environment development”. What kind of environment. We can have natural, social, economic, legal … etc. environment. Most probably you have a natural environment in mind. But what is its “development”? If there is the development, it should be some “goals” of this development. What are they? Maybe “natural environment state (or status, or condition)” would be better.
Response: Thank you for pointing all these out. I have revised all the ‘environment development’ into ‘environment’ including title, all the sections. Also I define the scope of our ‘environment ’in section 1 and 2, which has been highlighted in blue color. More details can be seen in my revised manuscript.
Page 3 – In my opinion, a letter x is not a good choice for variables. In many researches it is used as a symbol for random component. For your assumptions about random component you should add symmetry of its distribution (if you don’t want to have normality).
Response: Thank you for pointing all these out. Firstly, please accept my explanation about the ‘x’, which is an idiomatic usage from the perspective of statistical models. Thus I keep these to avoid too much letters make readers feel confusing. Secondly, I have revised the corresponding parts to illustrate assumptions about random component, which has been highlighted in blue color. More details can be seen in my revised manuscript.
Page 5 – There is something missing in the sentence starting with “Different from economy, …”
Response: Thank you for pointing all these out. I have deleted this sentence.
Page 5 – The IMD list of indicators shows that we are talking about “environmental protection” not “development”. How do you measure “sustainable development” (kind of magic key phrase) and “environmental laws”?
Response: Thank you for pointing all these out. I have revised all the ‘environment development’ into ‘environment’ including title, all the sections. About the “sustainable development” (kind of magic key phrase) and “environmental laws”, they are the indicators from IMD, which are the scores given by experts or institutes. I have added the measurement unit in Table 1 which helps readers to understand the indicators.
Page 7 – Hypotheses have not been “identified”. They were “formulated”
Response: Thank you for pointing all these out. I have revised the corresponding parts and highlighted in blue color.
Page 8, Table 1 – You should add “measuremet unit” column. Some of the variables are dummies (I suppose). How they were coded? What is “Government consumption”? When you use data from 61 economies – it should be expressed per capita (or somehow made comparable) and it should be stated in description or measurement column.
Response: Thank you for pointing all these out. I have added “measurement unit” column in Table 1 which helps readers to understand the indicators.
Page 11, Figure 3 – The figure is not clear. Everything should be explained, eg. ED1, ED2, ED3 – even just by refering to Table 1. Why arrows go from construct to variables when – if fact – the values constructs are “formed” by variables.
Response: Thank you for pointing all these out. I have added the explanation about all the indicators in Figure 3. About the arrows directions, I have explained the problem before Figure 3 and highlighted in blue color as follows.
It should be noted that, although the constructs cannot be directly observed and are formed by variables, the arrows still go from construct to variables which is consistent with the well-known European customer satisfaction index (Askariazad et al., 2015).
You should provide more arguments for usage of quantile regression, not just “basic” regression model.
Response: Thank you for pointing all these out. I have revised the corresponding parts and reorganized the structural of the paper, which has been highlighted in blue color.

Reviewer 4 Report
The format of references in the text is not suitable for sources that have more than two authors, the name of the first author should be mentioned only along with 'et al.'
Some references are not mentioned in the research text

Author Response
Dear reviewers and editors,
Thank you for your patience and kindness. The following are point-to-point responses respectively:
Comments and Suggestions for Authors
The format of references in the text is not suitable for sources that have more than two authors, the name of the first author should be mentioned only along with 'et al.'
Response: Thank you for pointing all these out. I have revised all the corresponding parts and highlighted in blue color.
Some references are not mentioned in the research text.
Response: Thank you for pointing all these out. I have deleted all the references which are not mentioned in the research text.

Round 2
Reviewer 2 Report
I have no further comments on the revision.
Reviewer 3 Report
Thank you for your detailed responses.